# Smart Meter Data Analysis of a Building Cluster for Heating Load Profile Quantification and Peak Load Shifting

**Yunbo Yang [1],\*, Rongling Li [1] and Tao Huang [2]** 

[1] Department of Civil Engineering, Technical University of Denmark, 2800 Lyngby, Denmark; liron@byg.dtu.dk
[2] Department of Engineering, Aarhus University, 8000 Aarhus C, Denmark; taoh@eng.au.dk
\* Correspondence: yunboy0829@gmail.com

**Abstract:** In recent years, many buildings have been fitted with smart meters, from which high-frequency energy data is available. However, extracting useful information efficiently has been imposed as a problem in utilizing these data. In this study, we analyzed district heating smart meter data from 61 buildings in Copenhagen, Denmark, focused on the peak load quantification in a building cluster and a case study on load shifting. The energy consumption data were clustered into three subsets concerning seasonal variation (winter, transition season, and summer), using the agglomerative hierarchical algorithm. The representative load profile obtained from clustering analysis were categorized by their profile features on the peak. The investigation of peak load shifting potentials was then conducted by quantifying peak load concerning their load profile types, which were indicated by the absolute peak power, the peak duration, and the sharpness of the peak. A numerical model was developed for a representative building, to determine peak shaving potentials. The model was calibrated and validated using the time-series measurements of two heating seasons. The heating load profiles of the buildings were classified into five types. The buildings with the hat shape peak type were in the majority during the winter and had the highest load shifting potential in the winter and transition season. The hat shape type's peak load accounted for 10.7% of the total heating loads in winter, and the morning peak type accounted for 12.6% of total heating loads in the transition season. The case study simulation showed that the morning peak load was reduced by about 70%, by modulating the supply water temperature setpoints based on weather compensation curves. The methods and procedures used in this study can be applied in other cases, for the data analysis of a large number of buildings and the investigation of peak loads.

**Keywords:** smart meter data; heat substation; clustering; load profile; peak load; load shifting

## 1. Introduction

Energy-saving and energy decarbonization have become part of energy strategies in many countries, as part of climate change mitigation efforts. In Demark, the electricity and heating should be 100% covered by renewable energy by 2035 [1]. The report launched by the Danish Commission on Climate Change Policy in 2010 stated that, by 2050, an energy system independent of fossil fuels is achievable in Denmark without high costs [2]. Such targets require imperative paradigm shifts to integrate energy systems and optimal operation on both energy supply and demand sides.

Buildings account for 40% of energy consumption in developed countries [3]. In Scandinavian countries, heating needs account for a large share of energy consumption. And in city area, district heating system (DH) cover 70–90% of the heating needs [4]. DH can play an essential role in the future

smart energy system, because of its flexibility in using various forms of energy sources [5]. The heating demand varies conspicuously in a day and consists of two parts, namely the baseload and the peak load. The heat in Greater Copenhagen (Copenhagen metropolitans) is currently produced by two types of heat generation plants owned by the utilities: baseload boilers and peak load boilers. The baseload boilers cover the continuous heat demand throughout the day. They are constituted by the waste incineration plants, geothermal plants, and biofuel based combined heat and power plants (CHP) in Greater Copenhagen. In comparison with the baseload, peak loads are fluctuating, and typically occur in the early morning between 06:00–09:00 [6] when occupants start their morning activities simultaneously in the residential buildings, and when the service systems in public/commercial buildings are rebooted. During the peak load hours, peak load boilers need to be used for heat production. However, the peak load boilers are mainly based on fossil fuels, such as natural gas and oil, which cause considerable $CO_2$ and particle emissions. On the demand side, high return water temperature from the buildings can cause penalties to the heat consumers in the building, if the cooling effect does not fulfill the requirements by the utility companies. Therefore, the benefit of shifting the peak loads is twofold: contributing to the decarbonization of DH and reducing the economic expenditure for heat consumers.

Load shifting can be achieved by reducing the peak load and shifting the peak demands to off-peak periods, thereby flattening the heat consumption profile. It can contribute to reducing the electricity bill by 20–25%, based on several techniques [7]. The studies of load shifting strategies in the electricity market are antecedent, and were summarized into three categories by Uddin et al. [8] Namely, the integration of energy storage system (ESS), demand side management (DSM), and the integration of electric vehicle (EV) to the grid. The former two strategies are also feasible to alleviate peak load issues in DH, and commonly supplement each other. The integration of ESS generally consists of two solutions. One is to implement additional heat storage units based on latent heat storage and thermochemical storage [9]. Christian et al. [10] investigated the potential to release power flexibility by utilizing thermal energy storage, such as different tanks in the power-to-heat system. However, this needs additional capital investment and is also limited by the availability of spaces. Another solution is to utilize the building thermal mass capacity, combined with the control strategies applied to the demand side [11]. It is shown that the thermal storage capacity of buildings can contribute considerably to the residential demand side management [12]. Favre et al. [13] addressed the importance of thermal mass in the role of load shifting by the optimal control of electrical heating systems. Implementing DSM through changing control strategies, for example, the optimal rescheduling of building heating systems [14,15], model predictive control (MPC) based on energy price [16] and efficiency of DH network [17]. In addition, the integration of heat pump coupled with DSM is also a promising approach for heating load shifting [18,19].

In recent years, conventional mechanical meters are being replaced by smart meters, from which real-time data regarding various attributes on both supply and demand sides are available. Data can be recorded in a short time-frequency (1–60 min), and transmitted to the service providers or consumers concurrently. This is of significance to enable utilities to monitor the status of energy networks and to provide researchers a new opportunity for a better understanding of consumer behavior [20], recognizing energy consumption patterns, detecting and diagnosing malfunctions, and further optimizing energy production and distribution [21], for example, using the statistical method and artificial neural network (ANN) to recognize energy patterns and automatically detect fault operations in a smart building cluster [22]. In Denmark, a large number of smart meters have been deployed in the energy network, especially in DH substations. It implies that hourly metering data for energy, water flowrate, and supply and return temperature are accessible. These data give us the possibility to extract information, such as energy consumption patterns, through data analysis. It is believed that, based on comprehensive data analysis, we can improve the design and operation of energy systems on both the supply and demand sides.

Machine learning is a commonly used data analysis technique. In general, machine learning methods can be divided into three categories, i.e., predictive method (supervised learning), descriptive method (unsupervised learning), and reinforcement learning [23]. Supervised machine learning mostly relies on the historical measured data, which are used to develop black-box models to predict future energy consumption (profiles), e.g., by using ANN [24] or applying a linear prediction model for MPC [25]. However, it requires a large amount of historical data to be adequately trained, and the results sometimes are absent with physical meaning [26].

Unlike supervised learning processes merely to predict the associated response between output with inputs, unsupervised learning pertains to discovering associated relationships between the inputs [27]. Unsupervised learning is often subjective and in need of beforehand domain knowledge. It can be classified into five categories: clustering, novelty detection, motif and discord detection, rule extraction, and visual analytics [28]. Among which, clustering analysis is rated as the most popular techniques, as it is less time-consuming and requires less human supervision [29]. Clustering has served as a powerful method to discover varied energy patterns from time-series energy consumption data [30,31], as it has been documented in [32] that the heating load has distinct daily and seasonal variation. Besides, the results of clustering can also be employed as pre-processed information for other analysis techniques. For example, Carmo et al. [33] and Gianniou et al. [34] used the clustered energy pattern for further logit regression analysis, to discover the factors influencing heat consumption in dwellings. Yang et al. [35] proposed a novel clustering algorithm that improves the accuracy of the prediction model by clustering more accurate energy patterns. Xue et al. [21] utilized the results of clustering to provide guidance for fault detection at DH substations.

The quantification of total peak loads of a building cluster and its corresponding load shifting potential, based on meter data, is still an area that needs more investigation. The present study aims to, first, analyze the smart meter data collected from DH substations of the typical municipal buildings in Copenhagen, using clustering analysis to identify typical load profiles and quantify peak loads; second, for typical load profiles with peaks, determine peak shifting potential using building simulations. As an example of the second aims, load shifting strategies are simulated using a model of a representative building developed based on time series measurements in two heating seasons, to show the possibilities to employ flexibility control in heat substations. The novelty of this study lies in: (1) categorizing heating load profiles base on the sharpness and duration of their peaks; and (2) quantifying the overall peak loads of a large number of buildings and their peak shaving potentials. In this paper, Section 2 demonstrates the data analysis procedures, the peak identification and quantification method, and the numerical model and simulation scenarios employed in this study. The results and discussions of the data analyses and the outputs of the simulations are described in Section 3. Conclusions are summarized in Section 4.

## 2. Methodology

The workflow of the load profile categorization and the investigation of peak load shifting potential based on a case study building is as follows:

(1) Data cleaning, including removing missing values and outliers and transforming the datasets into the appropriate form for the clustering analysis.

(2) Applying hierarchical clustering analysis to typical daily heating load profiles. Based on the clustering analysis results, we defined the peak load threshold to identify the peak load and categorize the load profiles based on the occurrence of the peak. Furthermore, peak quantification was conducted, based on the categorization of the load profiles. This is discussed in detail in Section 2.2, "Categorization of daily load pattern using clustering analysis, peak identification and peak load quantification".

(3) Finally, as an example, we investigated the load shifting potential of one representative building, using a numerical model developed based on the building time-series measurement data. Dymola

was used for modeling and simulation. R language was used for all data processing and analyses [36].

### 2.1. Description of Data

#### 2.1.1. Data Source—Description of DH Substations

The data used for the present study were collected from the DH substations of 61 municipal buildings located in Copenhagen, Denmark. The functions of the selected buildings are shown in Table 1.

**Table 1.** The distribution of the building functions.

| Building Function | Number |
| --- | --- |
| Daycare center | 38 |
| Inpatient institution | 10 |
| Teaching and researching facility | 7 |
| Others | 6 |

Heating substations are instantaneous heat exchanger systems connected to the DH network as the heat source for buildings. The substations consist of a primary side, i.e., the DH grid, and a secondary side, i.e., heating systems in buildings. Figure 1 shows a schematic view of a typical configuration of the DH substation. There are at least two loops on the secondary side, i.e., one or more than one loops for space heating (SH), depending on the building size, one loop for domestic hot water (DHW) preparation, and in some buildings, one or more loops for ventilation systems. All 61 substations are equipped with smart control systems, which target the secondary side supply water temperature as the control output by adjusting the water flow rate on the secondary side, according to a predefined weather compensation curve (WCC). The supply water temperature on the primary side is controlled by the utility, based on the regional heat demand and the weather of the day. It usually ranges from 60–110 °C during the year. In Greater Copenhagen, the DH supplies heat for DHW preparation all year round and for SH in heating seasons, which is usually from mid-September to mid-May of the next year (ca. 242 days). All the meters installed in the substation are smart meters, from which real-time data are recorded, with an hourly interval on both the primary side and the secondary side. The data can be accessed via the data management systems for the utility and the municipality.

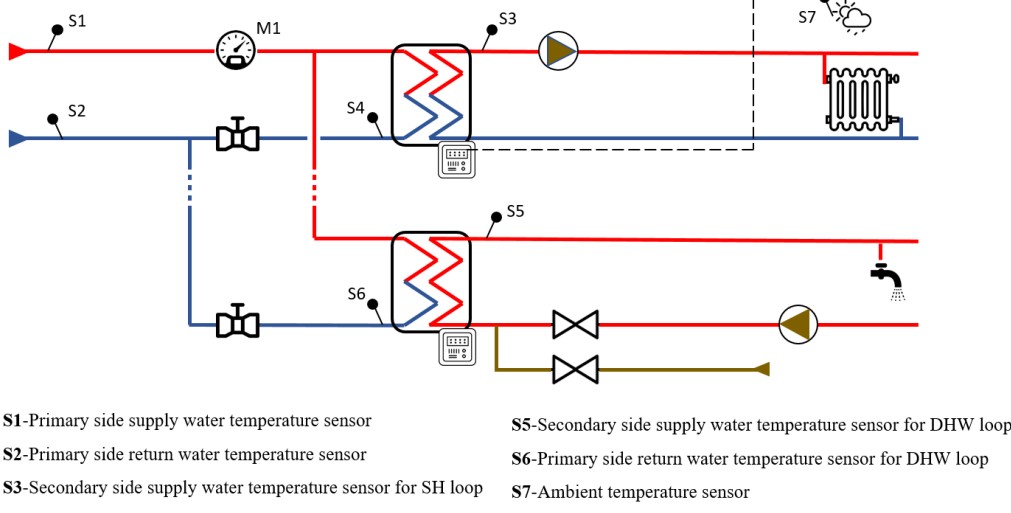

**S1**-Primary side supply water temperature sensor

**S2**-Primary side return water temperature sensor

**S3**-Secondary side supply water temperature sensor for SH loop

**S4**-Primary side return water temperature sensor for SH loop

**S5**-Secondary side supply water temperature sensor for DHW loop

**S6**-Primary side return water temperature sensor for DHW loop

**S7**-Ambient temperature sensor

**M1**-Primary side water flow rate meter

**Figure 1.** Schematic view of a typical substation, from Danfoss-ECL Comfort 310.

### 2.1.2. Dataset Characteristics

On the primary side, the data collection varied from one to two years. Table 2 lists the monitored attributes, which are identical in every substation.

**Table 2.** The measured attributes of data on the primary side.

| No. | Attribute | Unit |
|---|---|---|
| 1 | Datetime | yyyy-mm-dd hh:mm:ss |
| 2 | Energy | MWh |
| 3 | Cumulative flow | $m^3$ |
| 4 | Supply water temperature | °C |
| 5 | Return water temperature | °C |

On the secondary side, data are recorded and stored in the same format as those on the primary side. As all the smart control systems were installed in 2017, the measurement started on 1 July 2017, until the present. The monitored attributes on the secondary side are tabulated in Table 3.

**Table 3.** The measured attributes of data on the secondary side.

| No. | Attribute | Unit |
|---|---|---|
| 1 | Datetime | yyyy-mm-dd hh:mm:ss |
| 2 | Supply water temperature of loop $i$ | °C |
| 3 | Supply water temperature setpoint of loop $i$ | °C |
| 4 | Return water temperature of loop $i$ | °C |
| 5 | Return water temperature setpoint of loop $i$ | °C |
| 6 | Ambient temperature | °C |

### 2.2. Categorization of Daily Load Pattern Using Clustering Analysis, Peak Identification and Peak Load Quantification

#### 2.2.1. Categorization of Daily Load Pattern Using Clustering Analysis

The efficiency of clustering can be improved by data normalization, and the results of clustering strongly depend on the data normalization method, as normalization eliminates redundant data, and data on different scales. Z-Score normalization is commonly used in time series clustering analysis for energy consumption data [22,37]. The Z-Score standardizes the heat consumption data which is structured as each hour as a column and each day as a row. As a result, the normalized datasets are subject to standard normal distribution. The normalization is calculated with Equation (1):

$$z_i = \frac{x_i - \mu}{\sigma} \tag{1}$$

where, $x_i$ and $z_i$ denote the original value and the normalized/standardized value, $\mu$ denotes the mean of the dataset, and $\sigma$ denotes the standard deviation of the dataset.

The basic principle of clustering analysis is maximizing the dissimilarities between different clusters, while minimizing them within the same cluster. Normally, the pairwise distance or (dis)similarity is computed using a specific distance-based metric, and such a method strongly influences the goodness of the clustering quality. The commonly employed distance calculation methods include Euclidean distance, Manhattan distance, and correlation-based distances [38]. In this study, Euclidean distance is used, which is expressed in Equation (2).

$$d_{ecu}(x, y) = \sqrt{\sum_{i=1}^{n}(x_i - y_i)^2} \tag{2}$$

where, *x* and *y* are two data observations containing *n* attributes.

Hierarchical clustering is widely used among researchers in engineering. The most applied hierarchical clustering algorithm is agglomerative clustering, also known as *AGNES* (agglomerative nesting). This algorithm regards each observation as an individual cluster, and then merges the most similar clusters, until all clusters have been merged into one big cluster containing all observations. This strategy is denoted as the "bottom-up" approach [39]. Therefore, besides the distance measurement between observations, the (dis)similarity between clusters needs to be computed, to decide which clusters should be divided or merged. Agglomeration methods include complete linkage, single linkage, average linkage, centroid linkage, and Ward's minimum variance method. It is found that, for the present analysis, Ward's minimum variance method always yields the best results among others. It minimizes the total within-cluster variance, as at each step, the pair of clusters with minimum between-cluster distances are merged.

Hierarchical clustering was performed using the hourly heat demand data. Several tests for obtaining the optimal number of clusters were conducted. The goodness of clustering results was evaluated using the silhouette width and Dunn index [38]. It was found that clustering the dataset into three seasonal subsets always yielded the most rational cluster results. Therefore, in this study, all the datasets were clustered into three groups using the agglomerative hierarchical method, namely,

- winter season, from mid-November to mid-March;
- transition season, from mid-March to mid-May and mid-September to mid-November;
- summer season, from mid-May to mid-September.

### 2.2.2. Peak Identification and Quantification

In order to identify the peak from the load profile, a threshold of the heat load magnitude has to be defined. Different methods were used in previous studies to define the threshold, e.g., analyzing the local maxima/minima and define the threshold as the valley value before the highest maxima [40]; define the threshold by the difference between upper and lower loads [41]; using the values of the peak region relative to the mid-day load [42], etc. Zheng et al. [43] introduced the method to define different thresholds for different seasons. It is common in existing methods to use daily average energy load as the threshold to define peaks. However, using such methods, buildings without considerable peaks will also be included. Thus, in the current study, a new peak threshold ($P_{thd}$) is defined when the heating load exceeds the daily average heating power by 15%.

In order to perform the cross-comparisons among all the buildings, the identified peaks were further quantified, using two features to describe the shape of the peak load profile: the duration ($t_{peak}$), as shown in Figure 2 and the magnitude of the peak load ($\varepsilon$). Equation (3) expresses how $\varepsilon$ is calculated:

$$\varepsilon = \frac{P_{max} - P_{thx}}{P_{thd}} \tag{3}$$

where, $P_{thd}$ (kW) denotes the threshold for a heating load to be a peak load. $P_{max}$ denotes the maximum heating load.

### 2.3. Evaluation of Peak Load Shifting Potential

### 2.3.1. Model Description and the Case Building

A daycare center ($t_{peak}$ = 10 h, $\varepsilon$ = 13.5%) was selected for load shifting modeling. The building has three floors and a total heated floor area of approx. 900 m$^2$. It was constructed in 2005 and uses radiation system supplies by the district heating network. The material of the wall is brick, and the roof is covered by roofing felt. The exterior view of the building is shown in Figure 3. The secondary side of the substation consists of two SH loops and one DHW loop. The modeling of the DH substation was carried out in Modelica using a Dymola programming environment. The component models were selected from the LBL Buildings Libraries [44] and Modelica standards.

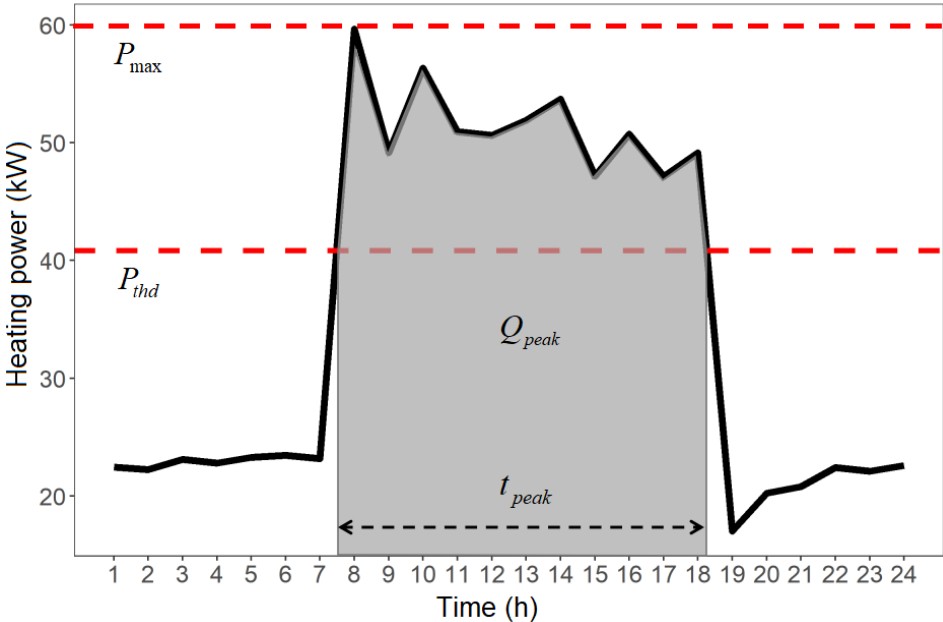

**Figure 2.** Illustration of parameters on an hourly heating load profile of a day.

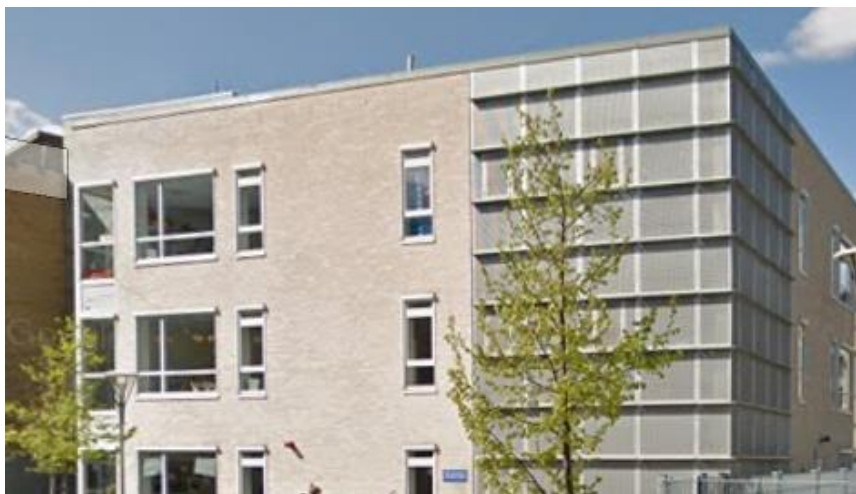

**Figure 3.** The exterior view of the case building for identifying the peak load shifting potential.

### 2.3.2. Model Calibration and Validation

The thermal zone model was calibrated using the 4-month measured data from November 2018 to February 2019. The calibration process consisted of two steps. First, the supply water temperature and the water flowrate from the real-time measurement data of SH and DHW loops, as well as the monitored outdoor temperature, were prescribed in the model as inputs. Then, the thermal conductance, the thermal capacity, the system nominal flow rate, and the heating power of the radiators were adjusted, until the simulated energy consumption data matched the measured values without compromising indoor thermal comfort. It was found that when the thermal conductance is 650 W/K, and the thermal capacity is 1255 kJ/K, the simulated energy consumption had a similar profile to the measurement data. Figure 4 shows an example of a 3-month comparison from December 2018 to February 2019, between the simulated results and the measurements.

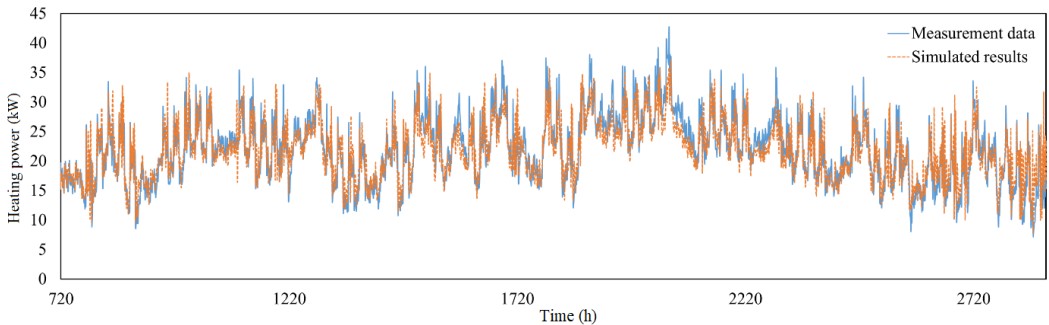

**Figure 4.** An example of a 3-month comparison between measured and simulated heat consumption after calibrating the thermal zone model.

The accuracy of the simulation method was measured by the coefficient of variation of the root mean squared distance (*CV(RMSD)*), see Equation (4),

$$CV(RMSD) = \frac{RMSD}{\overline{y}} = \frac{\left(\sum_{i=1}^{n}(y_i - \hat{y}_i)^2 / n\right)^{0.5}}{\overline{y}} \tag{4}$$

where, $y_i$ is the observed values from the measurements, $\hat{y}_i$ is the simulated values, $n$ is the number of variable observations and $\overline{y}$ is the mean of the observed time series. A lower *CV(RMSD)* indicates a higher accuracy of the simulation method and vice versa. Table 4 lists the *CV(RMSD)*. The low *CV* value indicates that the thermal zone model has been calibrated.

**Table 4.** *CV(RMSD)* of heat consumption in model calibration and validation.

| Model Calibration and Validation | Calibration of the Thermal Zone Model (4 Months) | Calibration of System Operation (4 Months) | Model Validation (5 Months) |
|---|---|---|---|
| *CV(RMSD)* | 0.126 | 0.198 | 0.204 |

In addition to the thermal zone, the system operation should also be calibrated. In this step, we used the measured supply water temperature setpoints and return water temperature setpoints as inputs, and use a P controller regulating the SH water flowrate. The system calibration was based on the calibrated thermal zone. The P controller was tuned until the simulated energy consumption profile matched the measured data, while the room temperature was maintained within the thermal comfort range. Figure 5 shows an example of a 3-month comparison of simulated and measured heat consumption from December 2018 to February 2019. The *CV(RMSD)* of this step is shown in Table 4.

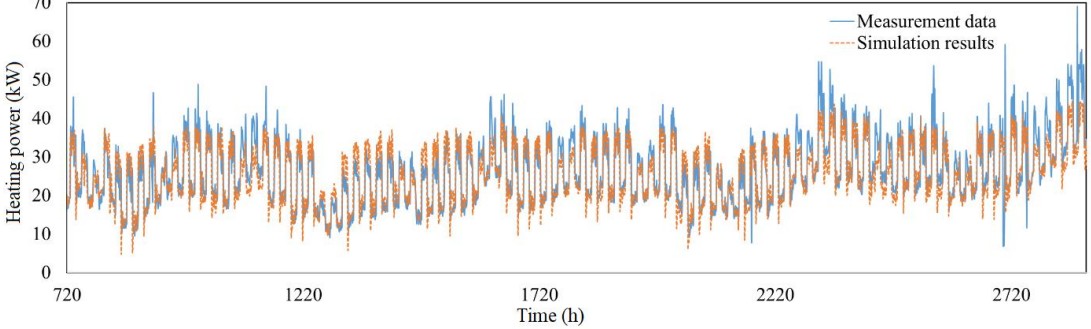

**Figure 5.** An example of a 3-month comparison between measured and simulated heat consumption, after calibrating the system operation.

For the validation of the model, 5-month measured data from November 2017 to March 2018 were used. Figure 6 shows a 3-month comparison between the measured and simulated heat consumption

from December 2017 to February 2018. The *CV(RMSD)* value in Table 4 indicates that the model can reproduce the actual energy consumption profile with a decent level of confidence.

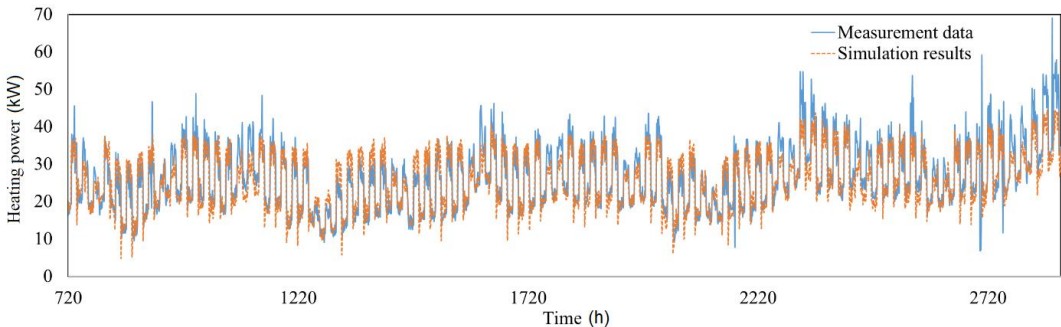

**Figure 6.** An example of a 3-month comparison between measured and simulated heat consumption in the model validation.

### 2.3.3. Load Shifting Scenarios

As all the investigated buildings are municipal buildings that rarely have evening peak loads, the focus of load shifting is thus on reducing morning peak loads occurring at 06:00–09:00. The supply water temperature for SH in the analyzed buildings is controlled by the predefined WCCs. The WCCs were obtained from the measurements and shown as the two reference curves in Figure 7 for the modeled building. To obtain these two separated curves from recorded data points (black dots in Figure 7), model-based clustering was applied. This method made the shape of the cluster explicit by assigning the probabilistic model to fit the data [45]. Subsequently, linear regression was applied to acquire the functions of two weather compensation curves, as shown in Figure 7. It was found from the dataset that night setback was applied in the building, and the two curves modulated the supply water temperature for 06:00–17:00 (working hours) and 18:00–05:00 (off-work hours), respectively.

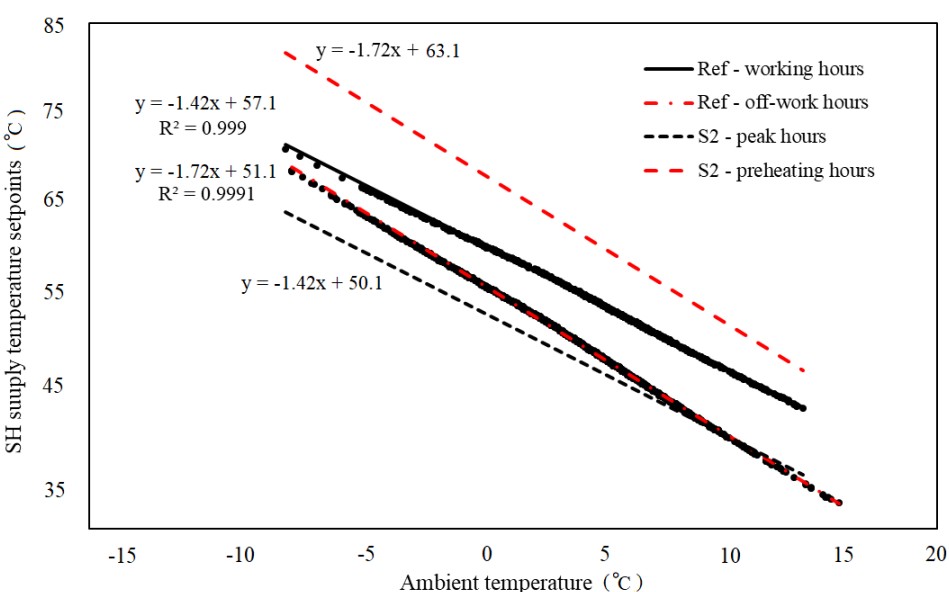

**Figure 7.** Weather compensation curves for the ref. case and scenario 2 (S2) as an example.

The morning peak load shifting can be achieved by preheating the building before the occurrence of the morning peak. In this study, the load shifting strategies were only applied and simulated to the SH circuit. The load shifting was carried out by increasing the supply water temperature setpoint and/or prolonging the preheating duration before 06:00, while reducing the setpoint during the morning peak hours without compromising indoor thermal comfort. Specifically, the slope and the intercept of

the two weather compensation curves were adjusted from the reference curves. To reduce the supply water setpoint at the same ambient temperature, we either decreased the intercept or increased the slope of the WCCs. Three scenarios were designed and simulated, with variable preheating setpoints, preheating periods, and morning setpoints, as shown in Table 5. The percentage of hours under thermal discomfort condition was maintained to be no more than 5% in all scenarios, according to DS/EN 15,251 Annex G for kindergarten Category II that defined the room temperature in the heating season as 20 ± 2.5 °C [46].

Scenario 1 (S1) was designed to reduce the morning (06:00–9:00) setpoints, while keeping the setpoints the same as the reference case during other hours. It utilized the night setback, which has already been implemented in the original control strategy. In S2 and S3, the setpoint was lowered during the morning peak hours (06:00–09:00), by reducing the intercept by 7 °C compared to the reference scenario. The variation between S2 and S3 was the preheating duration, as the preheating began at 04:00 for S2 and 03:00 for S3. The supply water temperature setpoints were higher in S2 than S3, to ensure the room temperature within a comparable range. Figure 7 shows the WCCs of S2 as an example. In all scenarios, during the period outside the preheating hours and peak hours, the setpoints were kept identical to the reference ones.

**Table 5.** Functions of weather compensation curves for all scenarios.

| Scenario | Period | Time | Function of Supply Temperature Setpoints for SH Loop $T_{2s,SH,sp}$ = |
|---|---|---|---|
| Ref. | Working | 06:00–17:00 | $-1.42 Tamb + 57.1$ |
|  | Off-work | 18:00–05:00 | $-1.72 Tamb + 51.1$ |
| S1 | Peak hours | 06:00–09:00 | $-1.43 Tamb + 51.8$ |
| S2, S3 | Peak hours | 06:00–09:00 | $-1.42 Tamb + 50.1$ |
| S2 | Preheating | 04:00–05:00 | $-1.72 Tamb + 63.1$ |
| S3 |  | 03:00–05:00 | $-1.72 Tamb + 61.1$ |

### 2.3.4. Performance Assessment

The effects on the load shifting of different scenarios were assessed using the indicators below, including the reduction of peak load ($\Delta Q_{peak}$) and the increment of energy consumption during preheating hours ($\Delta Q_{preh}$) in comparison to the reference case. They were used to describe the change of the energy consumption; load shifting factor (*LSF*) was used to measure the balance between the energy added for preheating the building and the energy reduced from the peak hours, as described in Equation (5); the morning peak load reduction rate in Equation (6) describes the reduction in energy use in morning peak hours in comparison to the reference case. Similarly, the load increasing rate in Equation (7) describes the extra energy use for preheating in preheating hours compared to the reference case; the increment of energy use at 10:00 (rebound effect) as to the reference case is described in Equation (8).

$$LSF = \frac{\left|\Delta Q_{peak}\right| - \Delta Q_{preh}}{\left|\Delta Q_{peak}\right| + \Delta Q_{preh}} \tag{5}$$

$$\text{Morning } Peak \text{ load reduction rate} = \frac{\Delta Q_{peak}}{Q_{peak,ref}} \times 100\% \tag{6}$$

$$Preheating \text{ load increasing rate} = \frac{\Delta Q_{preh}}{Q_{preh,ref}} \times 100\% \tag{7}$$

$$Rebound \text{ } effect = \frac{\Delta P_{10,S}}{P_{10,ref}} \times 100\% \tag{8}$$

where, $\Delta Q_{preh}$ denotes the difference of heat consumption during preheating hours between the individual scenarios and the reference; $\Delta Q_{peak}$ denotes the difference in heat consumption during peak hours (in this case study, this is at 06:00–09:00, between the individual scenarios and the

reference); $\Delta Q_{peak\_6-9}$ denotes the difference in peak load at 06:00–09:00, between the individual scenarios and the reference; $P_{10,ref}$ denote the heating power at 10:00 for the reference case; $\Delta P_{10,S}$ denotes the difference in the heating power at 10:00 between the individual scenarios and the reference. The subscript of *ref* is the corresponding absolute value of the reference case. *LSF* ranges between −1 and 1. If the additional energy added during the preheating period is identical to the reduced energy consumption during peak hours, *LSF* is 0. The extreme values of *LSF* = 1 means that morning heating load is shaved without preheating, and *LSF* = −1 means that there is no morning heating load reduction, even with preheating. The optimal operation being when $LSF \rightarrow 1$, namely the peak load is eliminated without preheating the building.

## 3. Results and Discussions

### 3.1. Clustering Results

For every building, the daily heating profiles were clustered into three clusters, which show distinct seasonality. As examples, the clustering result of the case study building is shown in Figure 8. For each cluster, a representative profile curve was formed by applying the median on the daily heating demand profiles in this cluster. Figure 9 presents three representative profiles of the three clusters, which are the median profiles of individual clusters. It is better to use median values than mode or mean values, because the underlying distribution is not normal [47]. Winter profile contained the coldest months of the year, with the highest heating power and an apparent variation between daytime and nighttime; in the cluster of the transition season, the heating power was much lower, and the variation between daytime and nighttime was not as noticeable as in winter. With the DHW demand as the only heat demand in the summer, the heating load was the lowest and flattest.

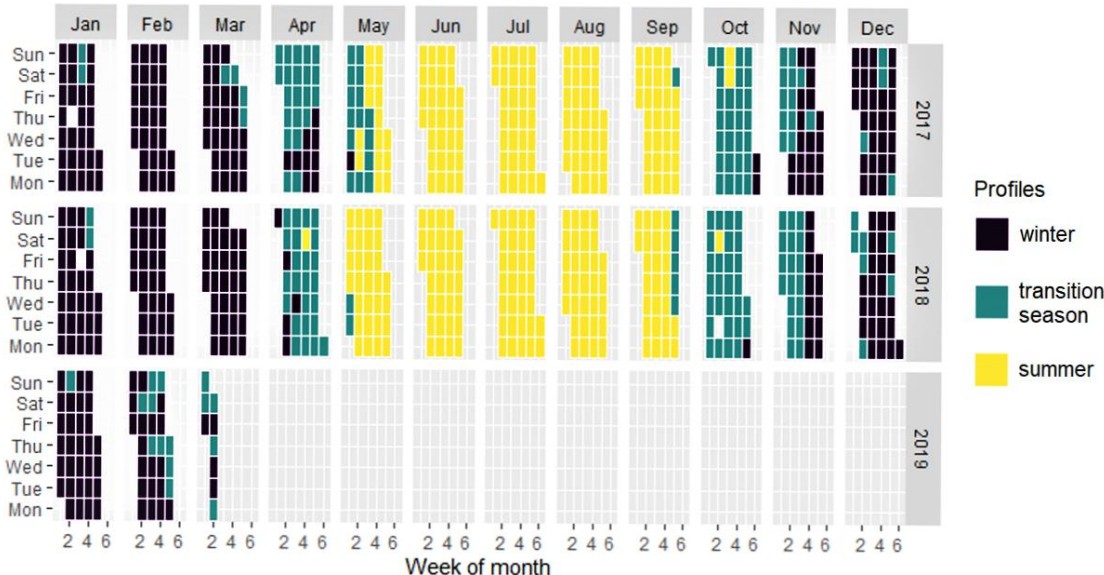

**Figure 8.** The distribution of three seasonal load profiles of the case study building.

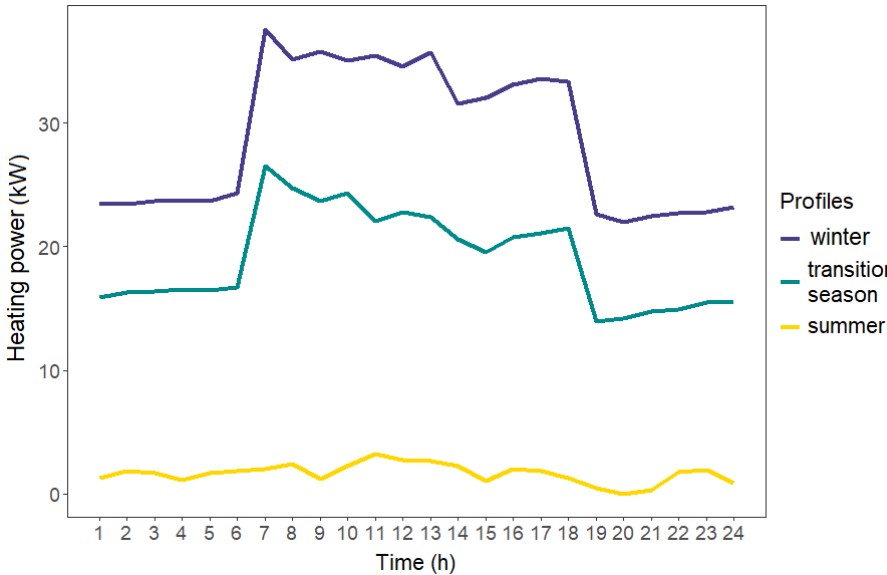

**Figure 9.** Representative seasonal heating load profiles of the case study building.

### 3.2. Results of Heating Load Profile Categorization and Peak Identification

For each building, the categorization of the heating load profiles was performed for winter and transition season, based on the $P_{thd}$ defined in Section 2.3.2. Overall, five peak profile types were classified for the winter and transition season among 61 buildings. Figure 10 presents the typical features of each type:

- Hat shape type, which has continuous high heat consumption during the daytime. The heat consumption during night and evening is stable and relatively low.
- Morning peak type, which has a noticeable short-term morning peak load occurring at 06:00-09:00. After that, the heat consumption reduces to a lower and stable level until the evening, when the substation is switched into night setback operation. The peak duration is much shorter compared with the hat shape.
- Morning and afternoon peak type, which has not only morning peaks, but also several additional peaks occurring in the afternoon. Typically, the highest peak is the first one in the morning.
- Morning and evening peak type, which has peaks during the evening period in addition to the morning peak.
- No peak, which has no peaks in the hourly heating profile of the day.

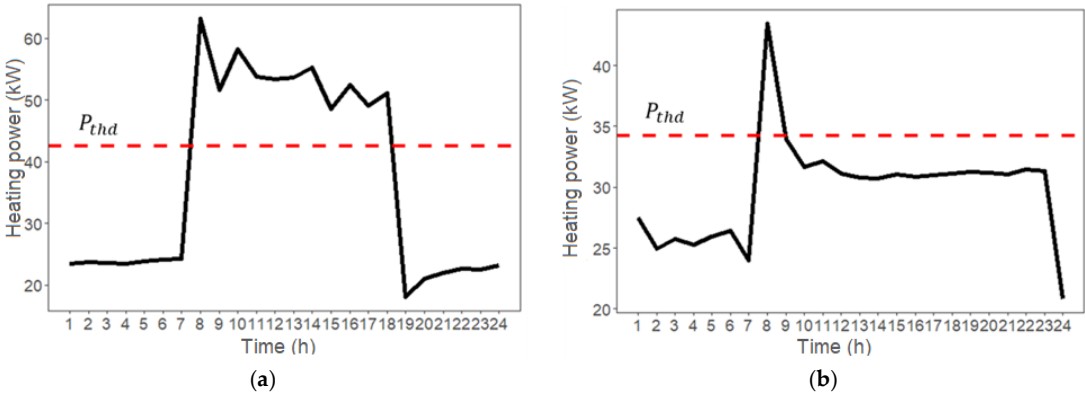

(a)  (b)

**Figure 10.** *Cont.*

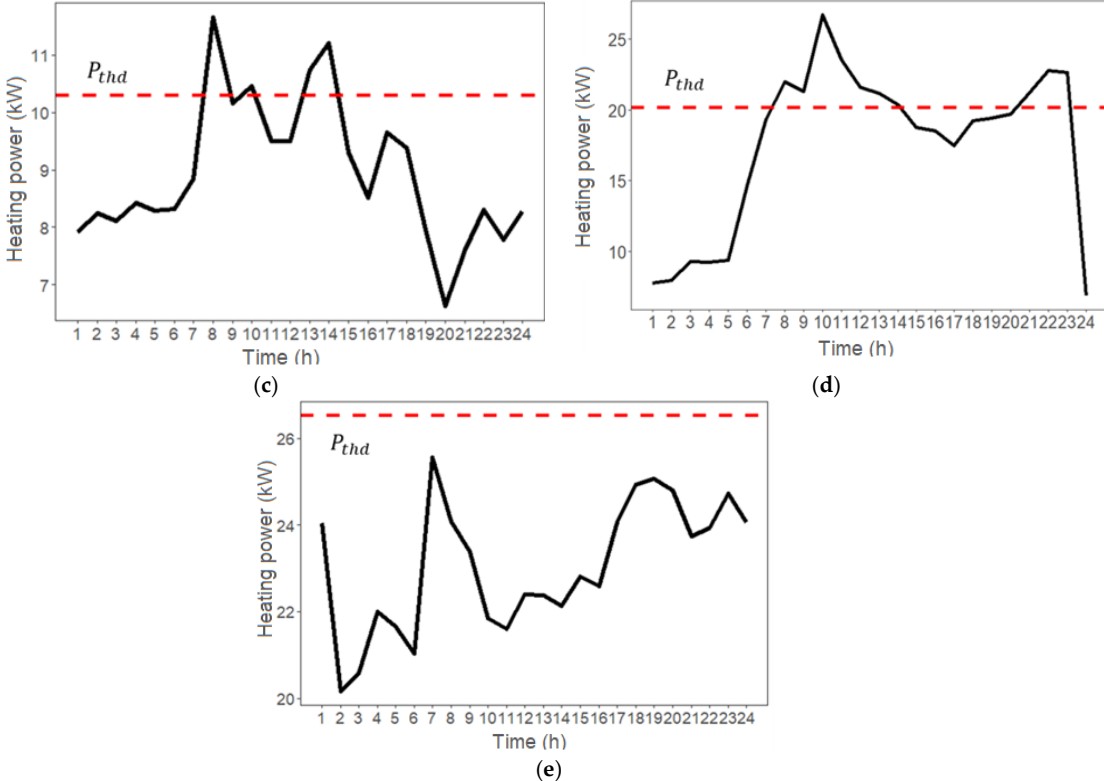

**Figure 10.** Typical hourly heating load profiles of each peak profile type. (**a**) Hat shape type; (**b**) Morning peak type; (**c**) Morning & afternoon peak type; (**d**) Morning & evening type; (**e**) No peak type.

Table 6 lists the distribution of the buildings in each peak type for the winter and transition season. In both the winter and transition season, most of the shape type lies in the morning peak. Hat shape and morning and afternoon type are the second-largest profile groups in winter and transition season, respectively. From the results, no correlation was found between the buildings' functionality and the peak type of the profile.

**Table 6.** Distribution of buildings of each peak profile type.

| Profile Type in Winter | Daycare | Nursing Home | Teaching | Others | Sum |
|---|---|---|---|---|---|
| Hat shape | 14 | 0 | 0 | 0 | 14 |
| Morning | 9 | 1 | 5 | 4 | 19 |
| Morning and afternoon | 10 | 2 | 0 | 0 | 12 |
| Morning and evening | 2 | 1 | 0 | 0 | 3 |
| None | 3 | 6 | 2 | 2 | 13 |
| Sum | 38 | 10 | 7 | 6 | 61 |
| **Profile Type in the Transition Season** | **Daycare** | **Nursing Home** | **Teaching** | **Others** | **Sum** |
| Hat shape | 3 | 0 | 0 | 0 | 3 |
| Morning | 20 | 7 | 7 | 5 | 39 |
| Morning and afternoon | 13 | 1 | 0 | 0 | 14 |
| Morning and evening | 1 | 1 | 0 | 0 | 2 |
| None | 1 | 1 | 0 | 1 | 3 |
| Sum | 38 | 10 | 7 | 6 | 61 |

### 3.3. Peak Load Quantification

Table 7 shows the results of the sum of peak loads for 61 buildings with different peak types in 2018. In the winter of 2018, the profiles with hat shape have the peak load ($Q_{peak}$) of 353.3 MWh and account for 43.7% of the total peak loads, and 10.7% of the total heating demand of all types ($\sum Q_{total}$).

The amount of peak load in the morning peak profile follows the hat shape with 200.2 MWh, and accounts for 6.1% of the total heating demand in the winter of 2018. In the transition season of 2018, the heating consumption during peak hours was dominated by the morning peak type of the profile by the value of 338.0 MWh, and accounts for 52.8% of the total peak loads and 12.6% of the total heating demand. The morning and afternoon profiles have similar peak loads in both winter and transition season at approx. 160 MWh. The ratio of the peak load over total heating demand of its own type ($Q_{total}$) can indicate the potential in terms of implementing load shifting strategies. From which, the buildings with hat shape profile type process a significant potential in both winter and transition season by a high ratio of approx. 55%. Morning and afternoon and morning and evening profile types also have considerable potential in applying load shifting strategies. Overall, the total peak loads were 808.0 MWh in the winter and 639.9 MWh in the transition season in 2018, and both account for approx. 24% of the total heating demand, respectively.

**Table 7.** Peak load quantification and the load sifting potential, based on categories in the winter and transition season of 2018.

| Profile Type in Winter | $Q_{peak}$ (MWh) | $Q_{total}$ (MWh) | $\frac{Q_{peak}}{Q_{total}}$ (%) | $\frac{Q_{peak}}{\sum Q_{total}}$ (%) |
|---|---|---|---|---|
| Hat shape | 353.3 | 642.7 | 55.0 | 10.7 |
| Morning | 200.2 | 1673.9 | 12.0 | 6.1 |
| Morning and afternoon | 167.5 | 687.8 | 24.4 | 5.1 |
| Morning and evening | 87.0 | 286.8 | 30.3 | 2.6 |
| Sum of all types | 808.0 | 3291.2 | - | 24.5 |
| **Profile Type in the Transition Season** | $Q_{peak}$ (MWh) | $Q_{total}$ (MWh) | $\frac{Q_{peak}}{Q_{total}}$ (%) | $\frac{Q_{peak}}{\sum Q_{total}}$ (%) |
| Hat shape | 73.8 | 136.1 | 54.2 | 2.8 |
| Morning | 338.0 | 1997.4 | 16.9 | 12.6 |
| Morning and afternoon | 156.7 | 384.1 | 40.8 | 5.9 |
| Morning and evening | 71.4 | 160.2 | 44.6 | 2.7 |
| Sum of all types | 639.9 | 2677.8 | - | 23.9 |

Figure 11 shows the distribution of peak types. In the winter, the bottom part is where the buildings with the morning peak type and the morning and afternoon peak type are distributed; the peak durations ($t_{peak}$) are less than 6 h and the peak sharpness ($\varepsilon$) is of a broad range from approx. 0% to 48%. At the right upper part, where the buildings with the hat shape peak type are distributed, $\varepsilon$ ranges from 10% to 48% and $t_{peak}$ is all above 6 h. It indicates that the hat shape peak typically has a sharper shape and longer duration. In the transition season, phenomena can be observed, as the buildings with the morning peak are still concentrated mainly at the left bottom part. In addition to these, the overall peak duration in the transition season is shorter, but $\varepsilon$ is larger compared with the winter. This is because the baseload in transition season is lower than that in the winter. Additionally, the peak load quantification results reveal that more attention should be paid to the hat shape type, especially in the winter. This is the reason that a daycare center with the hat shape peak ($t_{peak} = 10$ h, $\varepsilon = 13.5\%$) was selected for load shifting modeling.

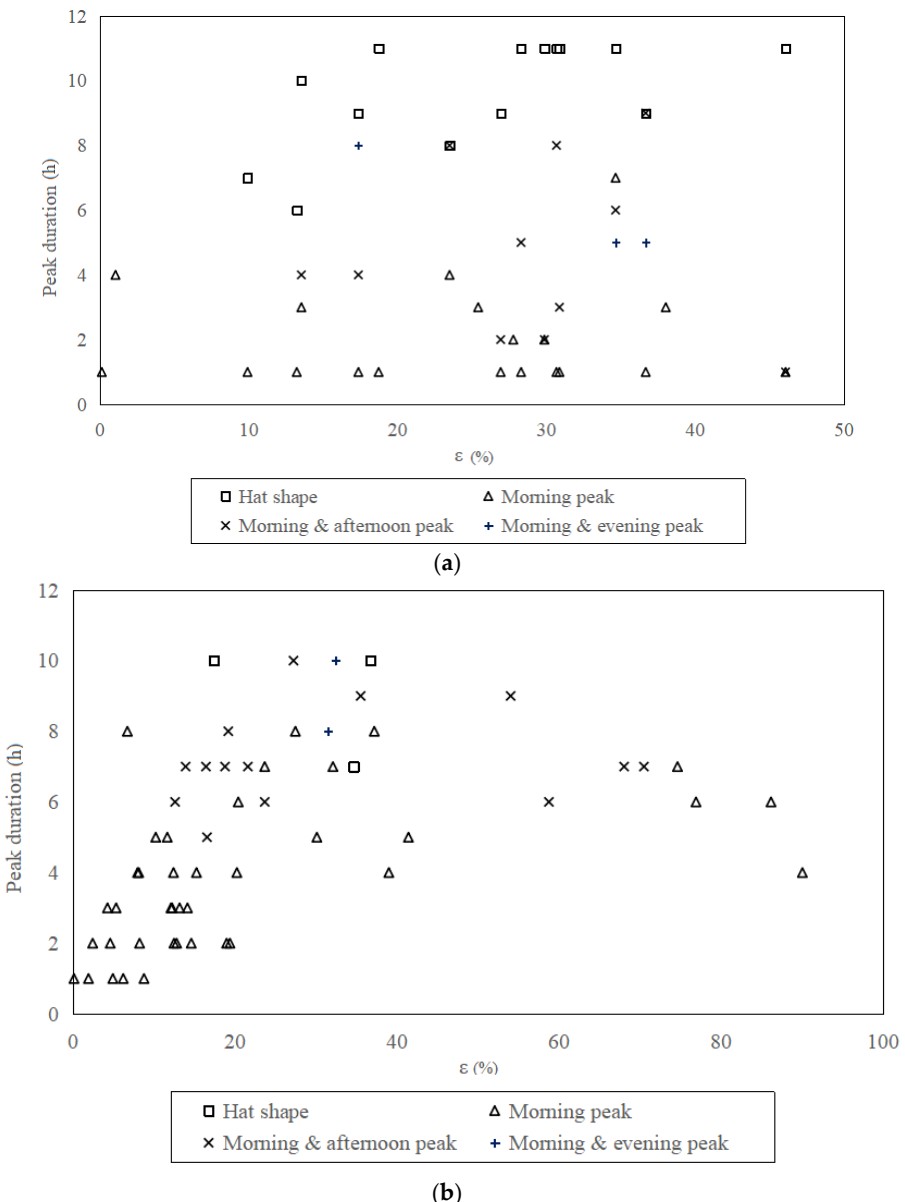

**Figure 11.** The distribution of peak types in winter and transition season. (**a**) The distribution in winter season. (**b**) The distribution in transition season.

### 3.4. Peak Load Shifting Potential

Table 8 shows the simulation results of three load shifting scenarios throughout the simulation period, from January 2018 to March 2018. Compared to the reference model, which has an average room temperature during working hours at 19.3 °C, all three scenarios sacrifice the room temperature for less than one degree, to the level of 18.5 °C, which still complies with the DS/EN 15,251 standard [46].

The morning peak load was shifted/reduced effectively, and the peak load reduction rate in morning peak hours was 68.7–70.2% in all scenarios in comparison to the reference, as shown in Table 8 Simulation results of load shifting scenarios during the winter of 2018. The rebound effect appeared in all scenarios when the temperature setpoints changed to normal after peak hours. In S1, without preheating, the peak load ($Q_{peak}$) was reduced by 68.7%, and the heating power was increased by 16% at 10:00 after the morning peak hours caused by the rebound effect. The morning peak load could be reduced by around 70% when preheating was implemented. S1 is preferable because of the highest *LSF* and a high peak load reduction rate.

Among the scenarios that applied with preheating strategies, S2 was of the highest *LSF* and highest morning peak reduction rate. The longer preheating period led to the smaller *LSF* because of a more considerable heat loss during the longer the preheating period. Compare S2 with S3; it can be observed that the increased preheating load is less "sharper" when the preheating period is longer since the setpoint for preheating is lower, which is indicated by the load increasing rate in preheating hours in Table 8. Overall, the energy consumption during morning peak hours for the reference scenario in 50 weekdays in winter 2018 was 4.27 MWh, and it could be reduced by 2.16 MWh by applying S1, and by 2.45 kWh by applying S2–S3.

**Table 8.** Simulation results of load shifting scenarios during the winter of 2018.

| Scenarios | Peak Load Reduced in Morning Peak Hours ($\Delta Q_{peak}$) (MWh) | Increased Load in Preheating Hours ($\Delta Q_{preh}$) (MWh) | Peak load Reduction Rate in Morning Peak Hours (%), Equation (6) | Load Increasing Rate in Preheating Hours (%), Equation (7) | Rebound Effect after Peak Hours (%), Equation (8) | *LSF* (-), Equation (5) |
|---|---|---|---|---|---|---|
| Ref. | 0 | 0 | 0.0 | 0.0 | 0.0 | - |
| S1 | 2.16 | 0 | −68.7 | 0.0 | 16.1 | 1.00 |
| S2 | 2.46 | 0.76 | −70.2 | 32.6 | 20.2 | 0.53 |
| S3 | 2.45 | 0.81 | −70.0 | 23.6 | 20.1 | 0.50 |

## 4. Conclusions and Discussion

Smart metering systems have been commonly deployed in DH networks and substations in Denmark, resulting in a large amount of data being available for analysis. In this study, the energy consumption data from the DH substations of 61 municipal buildings were analyzed using clustering analysis techniques. All the datasets were clustered into three clusters. The representative profiles obtained from clustering analysis were categorized by the peak occurrence of the profile curves. Furthermore, the peak loads were quantified using the duration and the sharpness of the peak with regard to different categorizations. To investigate the potential of peak load shifting, a numerical model of a representative building was developed, based on time-series measurements. The main conclusions are:

1. The energy consumption profiles were categorized into five types in the winter and transition season. Most of the buildings had the hat shape peak type during the winter and the morning peak type during the transition season. No correlation was found between the function of the buildings and the peak type.

2. For the analyzed 61 buildings, the total peak loads were 808.0 MWh in the winter and 639.9 MWh in the transition season in 2018. The hat shape type's peak load was predominating in the winter with 353.3 MWh, and accounted for 55% of the total peak loads. In the transition season, the morning peak type was the majority with a peak load of 338.0 MWh and accounted for 52.8% of the total peak loads. In the winter and transition season, the peak load accounted for 24% of total heating consumption. The buildings with the hat shape profile have been shown to have a notable potential for load shifting.

3. For the modeled building with a hat shape profile, the morning peak was reduced by about 70% in all scenarios, by implementing the proposed load shifting strategy. It was shown that the energy consumption during morning peak hours could be reduced by between 2.2 MWh and 2.5 MWh during the 50 weekdays in the winter. For the investigated building, the optimal start time of preheating was 2 h before the morning peak.

The results have shown that the seasonal operational patterns can be identified effectively using the clustering analysis technique. The median pattern of each cluster can be used to identify and

quantify peak loads. One limitation of this study is that only one building is modeled to test the effect of different load shifting scenarios. Therefore, the simulation result is too specific to be generalized. However, the proposed control strategy was shown to be effective for load shifting in this type of buildings. This strategy has the potential to be implemented in the existing DH substations by easily adjusting the weather compensation curves. It should be mentioned that the standard DS/EN 15,251 used in this study is replaced by DS/EN 16798-1:2019, which changes the indoor temperature recommendation (category II) from 17.5–22.5 °C to 20.0–24.0 °C. The load shifting results in this study are still valid regardless of this change, since the historical data used for modelling was collected during the time the previous standard was complied. In future studies, more load shifting strategies should be investigated in more building types with different energy consumption patterns, as well as the fact that the load shifting investigation should be based on the updated data and standard.

**Author Contributions:** Conceptualization, R.L., Y.Y. and T.H.; methodology, R.L., Y.Y. and T.H.; software, Y.Y. and T.H.; validation, R.L., Y.Y. and T.H.; formal analysis, Y.Y.; writing—original draft preparation, R.L., Y.Y. and T.H.; writing—review and editing, R.L., Y.Y. and T.H.; visualization, Y.Y. and T.H.; supervision, R.L.; project administration, R.L. All authors have read and agreed to the published version of the manuscript.

**Funding:** This research was funded by Innovation Fund Denmark, grant number: DSF1305-00027B.

**Acknowledgments:** This study was conducted in cooperation with Københavns Kommune. We would like to thank Theis Hybschmann Petersen for providing data and feedback on the analysis. This study is related to CITIES: Centre for IT-Intelligent Energy Systems in cities funded by Innovation Fund Denmark no DSF1305-00027B.

**Conflicts of Interest:** The authors declare no conflict of interest.

## Nomenclature

Indices, Parameters and Variables

| | |
|---|---|
| $x_i$ | origin value |
| $z_i$ | normalized value |
| $\mu$ | mean of dataset |
| $\sigma$ | standard deviation of dataset |
| $\varepsilon$ | indicator for peak load magnitude quantification |
| $P_{max}$ | maximum heating power of a load profile (kW) |
| $P_{thd}$ | threshold for peak load of a load profile (kW) |
| $P_{10}$ | heating power at 10:00 (kW) |
| $Q_{peak}$ | heating load during peak hours (kWh) |
| $Q_{total}$ | total heating load of the day (kWh) |
| $Q_{preh}$ | heating load during preheating hours (kWh) |
| $\Delta Q_{preh}$ | increased heating load during preheating hours in scenarios compared with the reference case (kWh) |
| $\Delta Q_{peak}$ | reduced heating load during morning peak hours in scenarios compared with the reference case (kWh) |
| Si | scenario i |
| $t_{peak}$ | peak load duration of a load profile (h) |
| $T_{2s,SH,sp}$ | secondary side space heating supply water temperature setpoints (°C) |
| $T_{amb}$ | ambient temperature (°C) |

## Abbreviations

| | |
|---|---|
| *CV(RMSD)* | Coefficient of variation of the root mean squared distance |
| DH | District heating |
| DSM | Demand side management |
| DHW | Domestic hot water |
| LSF | Load shifting factor |
| SH | Space heating |
| WCCs | Weather compensation curves |

## Subscripts

*ref*　　　　reference case
*S*　　　　scenarios

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
