# Peer review of "Smart Meter Data Analysis of a Building Cluster for Heating Load Profile Quantification and Peak Load Shifting"

_energies, doi:10.3390/en13174343_

Round 1

Reviewer 1 Report

Interesting paper, which shows a case study and its potential in the energy flexible buildings trend.

My concern is regarding the thermal environmental quality evaluation in the selected building. Authors use DS/EN 15251, which was replaced last year with DS/EN 16798-1:2019. Current recommendations (DS/CEN/TR 16798-2:2019) establish 20 °C as the minimum temperature for kindergartens (II cat.) during the heating season, with 20-24 °C temperature range for calculations of heating energy (19 °C is the minimum temperature for III cat.). Therefore, the peak shifting results in not meeting the recommendations by this building as the IV cat. is not recommended for kindergartens. This part of the paper should be updated and possibly more discussed.

Please read it again, thoughtfully, for correcting the spelling and punctuation. Over a dozen times, the punctuation mark is misused (e.g., comma or semicolon instead of a period, spaces are missing or doubled), "DN/ES 15251" instead of DS/EN 15251, etc. Moreover, please put attention to the formatting of tables - several of them are stretched over separate pages. The style also could be improved, e.g., rewrite some sentences to avoid dangling modifiers.

Author Response

Dear reviewer,

We greatly appreciate you for your complimentary comments and suggestions. We believe these have resulted in an improved revised manuscript. The manuscript has been revised to address the reviewer's comments. A point-by-point response is provided:

  1. Regarding the comment on the thermal environment recommendations, we think it is reasonable to use the regulation DS/EN 15251 since the data we analyzed is historical data collected when the old regulation has complied. The model was developed based on historical energy data. Your advice is definitely valuable for the analysis in the future when the data is updated. Therefore, we added a discussion to address this. Please see page 17, Line 471-476.
  2. We made changes throughout the manuscript. As these changes are a lot, we don’t list them here.

Reviewer 2 Report

The paper develops a very interesting topic and presents a methodology for monitor the status of heating load. Some comments are made to clarify this research and the results obtained:
1) I propose to extend the literature description with other methods of monitoring heating load,
2) I propose to broaden the description of the examined buildings (technology, material solutions, number of storeys, type of roof, type of wall insulation).

Author Response

Dear reviewer,

We greatly appreciate you for your complimentary comments and suggestions. We believe these have resulted in an improved revised manuscript. The manuscript has been revised to address the reviewer's comments. A point-by-point response is provided:

  1. We do not quite understand what is “the literature description with other methods of monitoring heating load”. The heating load is measured using smart meters at heat substations. In addition, the method of monitoring heating load is not the focus of this study, and the focus is on the analysis of historical data and modeling.
  2. More information on the case-study building is added in the text. Due to the limited accessibility to the building’s archive documents, we know only the material of the wall and roof, not their detailed thermal property. Please see page 7, line 236-239.